# Emerging Role of IGF-1 in Prostate Cancer: A Promising Biomarker and Therapeutic Target

**DOI:** 10.3390/cancers15041287

**Published:** 2023-02-17

**Authors:** Guoqiang Liu, Minggang Zhu, Mingrui Zhang, Feng Pan

**Affiliations:** Department of Urology, Union Hospital, Tongji Medical College, Huazhong University of Science and Technology, Wuhan 430022, China

**Keywords:** prostate cancer, IGF-1, IGF-1 receptor, biomarker, therapy

## Abstract

**Simple Summary:**

Prostate cancer (PCa) affects millions of men globally, and approximately 20% of PCa patients are found after they develop into a lethal metastatic or castration-resistant state. High levels of serum insulin-like growth factor-1 (IGF-1) and activated insulin-like growth factor-1 receptor (IGF-1R) in the prostate are found in PCa. A systematical understanding of the mechanisms of IGF-1 involved in PCa progression will expedite the development of new treatment approaches. The goal of this review is to summarize the literature on the role of IGF-1 in PCa to obtain a perspective regarding future scientific endeavors on PCa diagnosis and treatment.

**Abstract:**

Prostate cancer (PCa) is a highly heterogeneous disease driven by gene alterations and microenvironmental influences. Not only enhanced serum IGF-1 but also the activation of IGF-1R and its downstream signaling components has been increasingly recognized to have a vital driving role in the development of PCa. A better understanding of IGF-1/IGF-1R activity and regulation has therefore emerged as an important subject of PCa research. IGF-1/IGF-1R signaling affects diverse biological processes in cancer cells, including promoting survival and renewal, inducing migration and spread, and promoting resistance to radiation and castration. Consequently, inhibitory reagents targeting IGF-1/IGF-1R have been developed to limit cancer development. Multiple agents targeting IGF-1/IGF-1R signaling have shown effects against tumor growth in tumor xenograft models, but further verification of their effectiveness in PCa patients in clinical trials is still needed. Combining androgen deprivation therapy or cytotoxic chemotherapeutics with IGF-1R antagonists based on reliable predictive biomarkers and developing and applying novel agents may provide more desirable outcomes. This review will summarize the contribution of IGF-1 signaling to the development of PCa and highlight the relevance of this signaling axis in potential strategies for cancer therapy.

## 1. Introduction

Prostate cancer (PCa) is a highly heterogeneous disease harboring different mutations and tumor cell phenotypes that affects millions of men globally [1,2]. Most PCa cases are diagnosed at an early stage, with tumors limited to the prostate that grow slowly [2]. In approximately 20% of patients, tumors are found after they develop into a lethal metastatic state [3]. Moreover, advanced PCa cases often progress despite androgen ablation and are considered castration-resistant, which is a reason for the prominence of PCa as a cause of cancer-related death [4].

The oncogenesis of PCa is a multifactorial process associated with a complex interplay between inherent germline susceptibility, acquired somatic gene alterations, epigenetic modifications, and microenvironmental influences [1,5]. In solid tumors, the tumor microenvironment mainly consists of the tumor-associated stroma that includes infiltrating inflammatory and immune cells, resident fibroblasts, adipocytes, blood endothelial cells, and extracellular matrix (ECM) proteins [6]. Multiple soluble growth factors and cytokines mediate the intricate communication between malignant cells and surrounding environment cells to provide a niche for tumor growth and invasion [6].

Insulin-like growth factor-1 (IGF-1) is a single-chain peptide composed of 70 amino acids and shares 50% homology with insulin [7]. It contains three disulfide bridges, which create a tertiary structure that is critical for optimum binding to the insulin-like growth factor-1 receptor (IGF-1R). In normal individuals, IGF-1 can not only be delivered to target tissues by insulin-like growth factor binding proteins (IGFBPs) as a circulating hormone, but also synthesized in target organs, where it exerts actions through paracrine and autocrine mechanisms [8,9]. IGFBPs can bind approximately 98% of all circulating IGF-1 and form a trimeric complex with the acid-labile subunit (ALS) to serve as carrier proteins that regulate IGF-1 transport and prolong its comparatively short half-life [10]. Scholars have found six IGFBPs in our body, and approximately 80% of all bound IGF-1 are bound to IGFBP-3 [9,10,11]. In addition, the bioavailability of IGF-1 is negatively associated with the concentrations of specific IGFBPs in the extracellular fluids because IGFBPs have an even greater binding affinity for IGF-1 than IGF-1R [9,10].

IGF-1R is widely displayed on the surface of normal tissue and solid tumor cells [10,12]. IGF-1R is composed of two extracellular α-subunits that are activated upon IGF-1 binding and two β-subunits that have intracellular tyrosine kinase domains that are phosphorylated by IGF-1. Activated IGF-1R can activate phosphatidylinositol-3 kinase (PI3K)/serine-threonine kinase (Akt)/mammalian target of rapamycin (mTOR) and Ras/Raf/mitogen-activated protein kinase (MAPK) signaling to achieve cell survival and proliferation [13]. It can also undergo internalization and translocate to the nucleus of cells, where it can modulate the expression of genes involved in cell cycle regulation, DNA synthesis, and damage repair [14,15,16]. In summary, IGF-1 signaling can enhance cell growth due to its anabolic effects [10].

In recent decades, multiple studies have clearly shown that IGF-1 signaling is closely related to PCa development and progression. Based on previous findings, we will focus on the correlation of IGF-1 signaling with the risk of PCa, propensity for metastasis, therapy resistance, and cancer-related death in this review. The associated mechanisms and several agents devoted to improving the survival of PCa patients by interfering with the IGF-1/IGF-1R axis that are being tested in clinical trials and preclinical experiments will also be described. This review will provide a detailed overview outlining the overall effects of IGF-1/IGF-1 receptor signaling in PCa, enabling additional in-depth research and discussion.

## 2. Circulating IGF-1 Levels and Local IGF-1R Expression Are Related to PCa Development

The role of IGF-1 signaling in controlling rates of cell renewal has led to an interest in the relevance of this regulatory system to neoplasia [17]. Emerging results have indicated that high levels of circulating IGF-1 and activated IGF-1R are associated with an increased risk of progression in several solid tumors, such as breast cancer and PCa [9]. The association between serum IGF-1 or tumor IGF-1R activity and PCa has been reported (summarized in Table 1) [18,19,20].

### 2.1. Tumor Initiation: Risk of Tumorigenesis

Insulin growth factors are potent mitogens that regulate cell survival and cell proliferation and are involved in PCa development [21]. The median molar IGF-1: IGFBP ratio was found to be related to the incidence of PCa, with reduced levels of IGFBPs or elevated levels of IGF-1 being related to an elevated risk of PCa [22,23,24]. A prospective study reported that men with serum IGF-1 values in the highest quartile had an approximately 4.5 times greater risk than men with serum IGF-1 values in the lowest quartile, even among men with a baseline PSA level ≤ 4 ng/mL [20]. These results suggest that IGF-1 may be a significant independent predictor of PCa risk [20].

In addition to IGF-1 and its binding proteins, IGF-1R activity is also elevated in PCa in comparison to benign prostate tissue, and this increased activity commonly persists in metastatic PCa [25,26,27,28,29]. IGF-1R is predominantly membrane-localized in benign glands, while malignant epithelium contains prominent internalized IGF-1R (located in the nucleus and cytoplasm). IGF-1R can be phosphorylated by its ligand or autophosphorylated, subsequently being rapidly internalized to mediate IGF-1-induced signaling that promotes tumor cell growth and proliferation [30]. Therefore, IGF-1R, especially internalized (i.e., “after phosphorylation”) IGF-1R expression in the prostate, is also a novel predictor of PCa risk.

### 2.2. Tumor Progression: Cell Migration and Metastatic Spread

There have been contradictory results regarding the relationship between PCa tumor grade and circulating IGF-1 levels from different studies. One large case-control study with 1260 subjects illustrated that high serum IGF-1 levels were significantly associated with an increased risk of PCa progression [19]. In addition, PCa cells that had metastasized to bone showed an upregulated IGF-1 regulatory system, indicating that IGF-1 may promote cancer cell metastatic spread. However, other studies showed that there was no correlation between serum IGF-1 levels and serum PSA levels, Gleason score, or positive biopsy cores [31,32]. Furthermore, another study showed that serum IGF-1 levels, but not PSA levels, tended to be lower in patients with higher surgical Gleason scores. This result may explain why high-grade PCa is less sensitive to the action of IGF-1 than low-grade PCa [33]. Circulating IGF-1 has stronger effects on the development of low-grade PCa than on the development of high-grade disease [34]. Differences in study subjects or the duration of follow-up between studies may explain the inconsistency in the relationship between tumor grade and IGF-1 level.

In contrast to serum IGF-1, the intraprostatic expression of IGF-1 has a more consistent association with tumor grade, pathological stage, and disease progression. Intraprostatic expression of IGF-1 could be more significant than circulating levels of growth factors [31]. In a transgenic adenocarcinoma of mouse prostate (TRAMP) model, prostate-specific IGF-1 mRNA expression was increased during PCa progression and in the accompanying metastatic lesions [35].

Enhanced nuclear and Golgi-associated IGF-1R signaling are also features of preinvasive lesions and invasive cancers. They are significantly associated with advanced tumor stage in patients suffering from PCa. Impaired nuclear and Golgi IGF-1 signaling inhibits growth factor-induced cancer cell migration [30,36].

Overall, both increased IGF-1 levels and activated IGF-1R signaling are correlated with PCa progression. Moreover, the levels present in cancer cells or their surroundings are more relevant than the level in the organism as a whole.

### 2.3. Biological Tumor Recurrence

Biological recurrence after radical intervention, including surgery or radiation therapy, is one of the main issues in the treatment of PCa. Given the heterogeneous nature of this disease, there is no infallible method of distinguishing aggressive from indolent tumors and no single biomarker predictive of PCa outcome [3,13]. Currently, clinicians rely on serum PSA and a combination of clinical and pathological variable-based prediction models to identify patients who are likely to experience recurrence [13].

A portion of studies have reported that serum IGF-1 level is a risk factor for predicting recurrence after surgery or drug therapy in breast cancer [37], hepatocellular carcinoma [38], and gastric cancer [39]. The risk of PCa biological recurrence predicted based on serum IGF-1 level may not be accurate due to the relatively slow progression of PCa and limited follow-up time of PCa studies [13,40]. Whether serum IGF-1 can be used as a marker for predicting the risk of recurrence remains controversial. In contrast to serum IGF-1, IGF-1R activity, especially cytoplasmic and nuclear IGF-1R expression, has been proven to promote radiotherapy resistance and enhance the risk of biological cancer recurrence [41]. Other studies have demonstrated that internalized IGF-1R is an independent predictor of biological recurrence and may be a better biomarker than total IGF-1R [36,42].

**Table 1 cancers-15-01287-t001:** Population case-control analysis of serum IGF-1 and local IGF-1R expression in PCa.

Study Population	Region	PCa Risk/Stage/Survival Related to IGF-1 or IGF-1R Expression	Main Conclusion	References
152 PCa cases and 152 control subjects (median age = 60)	United States	PCa risk has a strong, positive association with IGF-1 levels (RR = 4.3); The increased risk associated with IGF-1 was stronger among the older men (age ≥ 60); No significant difference in IGF-1 for high-grade/stage and for low-grade/stage cancers.	IGF-1 is a significant independent predictor of PCa risk; PSA has a weak, positive association with IGF-1 levels.	[20] (Science,1998)
630 PCa cases and 630 control subjects (median age = 65)	Europea	Serum IGF-1 concentration is mildly associated with PCa risk (OR = 1.39); The association of serum IGF-1 concentration with risk was slightly stronger for advanced-stage disease (OR = 1.76).	A weak positive association between IGF-1 concentration and overall PCa risk.	[19] (Cancer Epidemiol Biomarkers Prev, 2007)
1709 PCa cases and 1778 control subjects	United States	Plasma IGF-1 is associated with increased risk of PCa (OR = 1.28); Higher IGF-1 was more significantly positively associated with risk of low-grade PCa, but not intermediate- or high-grade PCa.	Being high in IGF-1 and can elevate the risk of PCa.	[22] (Int J Cancer, 2015)
793 PCa cases underwent radical prostatectomy and 272 men with negative prostate biopsy (mean age = 65)	Korea	High serum IGF-1 was associated with a high risk of localized prostate cancer (OR = 3.35) but not the risk of advanced pathologic stage (*p* = 0.911); High serum IGF-1 levels are associated with a low risk of high surgical GS (OR = 0.464); Serum IGF-1 levels are significantly correlated with serum bioavailable testosterone levels (*r* = 0.157).	Serum IGF-1 may represent a valuable marker of surgical GS.	[34] (Cancer Med, 2018)
156 PCa cases (median age = 67) and 271 control subjects (median age = 69)	Austria	Serum levels of IGF-1 have no correlation to serum PSA, Gleason score, and number of positive biopsy cores.	Quantification of IGF-1 levels may not provide useful information in the diagnosis of PCa.	[32] (European Urology, 2005)
72 PCa cases and 50 control subjects (median age = 67)	Unknown	Patients with higher IGFBP-1 levels have a shorter time to the development of CRPC; A higher serum IGF level in itself does not seem to adversely affect the time to CRPC.	Elevated IGFBP-1 seems to be associated with shorter time to CRPC and lower overall survival in men with metastatic PCa.	[23] (Prostate, 2014)
753 PCa cases in various stages (median age = 66)	United Kingdom	IGF1R expression is significantly higher in tumor tissue compared with normal-appearing tissue but not GS or pathologic/clinical tumor stage; Strong IGF-1R expression is associated with a borderline significantly increased risk of lethal PCa (HR = 1.7).	IGF signaling in prostate tumors plays a role in the progression of prostate cancer.	[28] (Carcinogenesis, 2018)
360 patients underwent surgery for PCa or BPH (median age = 68)	Japan	IGF-1R was more expressed in patients with PCa compared to the BPH (100% vs. 0%); IGF-1R positivity was higher in Ki-67 + (78.6% vs. 45.5%) in INS R-α + (84.4% vs. 59.8%) and INSR-β + (9.4% vs. 1.5%) tissues.	IGF-1R is associated with greater tumor aggressiveness in PCa patients with diabetes.	[29] (Translational Research, 2021)
130 patients with PCa (median age = 63)	United Kingdom	IGF-1R expression significant increase in cancerous versus benign tissue (*p* < 0.001); No significant difference in IGF-1 levels and IGF-1R expression between non-BCR and BCR patients (*p* = 0.576 or 0.149).	Tissue proteins (PTEN/ INSR/IGF-1R) may help patients select postoperative adjuvant therapy and prevent BCR.	[13] (Prostate, 2017)
136 patients with PCa (median age = 69)	Japan	Higher-grade tumors (primary GS = 4–5) contain significantly more IGF-1R than lower-grade tumors (primary GS = 3) (*p* = 0.004); The risk of overall recurrence was significantly greater in men whose prostate cancers contained high total- and cytoplasmic- IGF-1R (*p* = 0.002).	Evaluation of IGF-1 inhibition as a novel route to radiosensitization of prostate cancers that express high total- or cytoplasmic- IGF-1R.	[42] (British Journal of Cancer, 2017)
215 patients of PCa with bone metastasis (median age = 70) or 111 patients of PCa with bone metastasis (median age = 70.6)	United Kingdom	Cancer-specific survival was significantly associated with the CA repeat polymorphism (*p* = 0.013); Patients with at least one C-T haplotype showed significantly worse survival compared with those who had no C-T haplotype (*p* = 0.0003) The median survival time was 41 months and 61 months for patients with and without the long allele of the IGF-1 polymorphism, respectively (*p* = 0.019);	Polymorphisms of the IGF-1 especially C-T haplotype are associated with worse survival of PCa patients with bone metastasis at initial diagnosis and may be a novel predictor in PCa patients with bone metastasis.	[3,43] (BMC Cancer, 2013) (J Clin Oncol, 2006)

Abbreviations: RR, relative risk; OR, odds ratio; PCa, prostate cancer; CRPC, castration-resistant prostate cancer; BPH, benign prostatic hyperplasia; IGF-1R, insulin-like growth factor-1 receptor; INSR-β, insulin receptor-β; GS, Gleason score; BCR, biochemical recurrence; PTEN, gene of phosphate and tensin homolog deleted on chromosome ten.

### 2.4. Cancer-Specific Survival

The prognosis for an individual with PCa is highly variable and dependent on tumor grade at primary diagnosis [5]. Resistance to hormone therapy and androgen-independent growth of cancer cells are considered critical factors affecting patient survival [3].

Given that the growth and proliferation of various cancer cells can be promoted by IGF-1, the feasibility of using IGF-1 as a marker of the prognosis of PCa was determined. A study published that the number of the *IGF-1* (*cytosine-adenine (CA)*) repeat polymorphism is a risk factor for PCa and an independent predictor of survival [3,43]. The *CA* repeat polymorphism is located in the promoter region of the *IGF-1* gene, and it was found to be significantly associated with higher IGF-1 levels in circulation and worse cancer-specific survival in PCa [43]. Additionally, nuclear localization of the insulin-like growth factor-1 receptor (nIGF-1R) in tumor cells is emerging as a potentially vital factor in tumor pathophysiology and has been linked to adverse clinical outcomes in various cancers [44,45].

## 3. The Mechanism by Which IGF-1 Signaling Regulates PCa Progression

In PCa, IGF-1 is synthesized and secreted from both cancer cells and the local niche and functions in an autocrine or paracrine manner [46]. IGF-1 induces phosphorylation and internalization of its receptor and thereby induces multiple biological changes that facilitate tumorigenesis, cancer progression, metastatic spread, and even treatment resistance. The molecular mechanisms by which IGF-1/IGF-1R signaling meticulously regulates PCa development and progression have been investigated in detail (shown in Figure 1 and Figure 2).

### 3.1. Cancer Cell Survival and Renewal

An intricate balance between cancer cell proliferation-related elements and apoptosis-regulating elements is critical for preventing PCa growth. Disruption of the balance between these elements triggers evasion of apoptosis and promotes cell survival, thus contributing to cancer initiation and progression [33]. As a factor affecting this balance, IGF-1 signaling is essential for the survival and proliferation of many normal and malignant cell types and protects these cells from programmed cell death [47,48]. Loss of IGF-1R induced by transient transfection with small-interfering RNA (siRNA) oligonucleotides or inhibition of IGF-1R activity by specific inhibitors inhibits the survival and proliferation of PCa cells [49,50,51]. Several mediators have been reported to participate in the regulation of cell survival and proliferation triggered by IGF-1R signaling, including forkhead box transcription factors (FOXOs), oncogenes, and tumor suppressor genes, as elaborated in the following paragraphs.

The binding of IGF-1 to its tyrosine kinase receptor results in the phosphorylation of several cellular proteins, including FOXO transcription factors, Bcl-2-associated agonist of cell death (BAD), and glycogen synthase kinase-3 (GSK3α/β), to facilitate cell survival and cell cycle entry via phosphorylation of Akt on Thr308 or Ser253 [52,53]. For instance, activated IGF-1R signaling dampens FOXO3 signaling and thus reduces the expression of the proapoptotic protein Bim to inhibit tumor cell mitochondrial apoptosis by increasing the activity of the phosphorylated PI3K-Akt pathway (Figure 2) [54]. As one of the most highly expressed members of the FOXO family in PCa cells, FOXO3 expression is negatively correlated with PCa progression [55]. While FOXO3 is active inside the nucleus, nuclear exclusion and accumulation of FOXO3 in the cytoplasm are induced by AKT-induced phosphorylation [53]. Therefore, the level of nuclear FOXO3 is decreased, which prevents FOXO3 from playing an anticancer role and mediates the promotion of adenocarcinoma growth by IGF-1 [54].

IGF-1R can enhance the expression of cyclin D1 and the progression of the cell cycle from the G1 to the S phase through three mechanisms (Figure 2). First and foremost, the Ras/MAPK pathway is one of the main downstream mediators of IGF-1 signaling and is closely connected with cell cycle transitions and cell proliferation. IGF-1 can induce tyrosine phosphorylation of β-catenin and promote its dissociation from a complex and transport it into the cytoplasm. In cancer cells, β-catenin accumulates in the cytoplasm, translocates into the nucleus, and interacts with T-cell factor/lymphoid enhancer factors (TCF/LEF) to activate cyclin D1 and promote cell cycle transitions [56]. Finally, SUMOylated IGF-1R undergoes nuclear translocation and binds to enhancers or nuclear proteins to activate cell cycle-regulated genes and increase G1-S progression [44].

In addition to the mechanisms illustrated above, several studies have reported that IGF-1 signaling also mediates cell survival and proliferation by disturbing the expression of oncogenes and tumor suppressor genes. For example, internalized IGF-1R can upregulate the expression of the oncogene *JUN* and enhance the recruitment of RNA Pol II to *JUN* promoter regions, which is beneficial to cell survival (Figure 2) [36]. In addition, increased IGF-1R activity inhibits PTEN expression, which is a critical regulator of the suppression of prostate epithelial cell proliferation, thus strengthening prosurvival and proliferation signals [21,57].

### 3.2. Cancer Cell Migration and Metastasis

The metastatic spread of cancer cells is a complex process that requires dynamic crosstalk between cancer cells, vascular endothelial cells, cell adhesion molecules, and ECM substrates [51]. Activated IGF-1/IGF-1R signaling likely stimulates PCa cell dissemination to other organs by modifying cell adhesion and mobility, facilitating angiogenesis, and increasing the osteoblastic activity of neoplastic cells (Figure 1) [58].

#### 3.2.1. Cell Adhesion and Motility

Cancer cell invasion is closely connected with cell adhesion and motility, which are largely mediated by adhesion-related proteins [56,59]. Disturbed cell adhesion and enhanced cell motility contribute to aggressive cancer behaviors. Both E-cadherin and β-integrin are common adhesion-related proteins that can regulate IGF-1/IGF-1R signaling to facilitate cell migration [60,61]. While E-cadherin is a notable factor responsible for enhancing IGF-1R stability and plasma membrane localization in cells with low migratory ability and even nonmigratory cells, β-integrin is a meaningful factor for determining IGF-1R translocation to or release from the Golgi apparatus in cancer cells, and its subcellular localization is associated with cancer cell migratory potential and aggressive behaviors [30,62].

Activated IGF-1R is commonly present at focal adhesions, where it can form a ternary complex containing integrin/IGF-1/IGF-1R and protect β-integrin from proteasomal or lysosomal degradation, thereby affecting the expression of β-integrin and inhibiting cell adhesion to the basement membrane in tumors [50,51]. Loss of IGF-1R ameliorates the tumor-promoting effects due to enhanced proteasomal degradation of β-integrin subunits [49,50]. Two tyrosines (Tyr1250 and Tyr1251) in the C-terminus of IGF-1R are considered to be essential for integrating IGF-1R and adhesion signaling [63]. Mutation of these two tyrosines is sufficient to abrogate IGF-1R’s function in facilitating cellular transformation and migration [30,64]. The interplay between IGF-1R and cell adhesion molecules collectively activates Akt and further signaling pathways to activate the mitotic cascade, thus contributing to the development and aggressiveness of cancers [51]. In addition, IGF-1R can also regulate cell adhesion by influencing matrix metalloproteinase (MMP) expression, which is essential in the degradation of ECM components, including gelatin and collagens, and plays a crucial role in helping single cancer cells to break away from primary tumors and blood vessels during cancer metastasis (Figure 1) [65,66].

Moreover, the adhesion and motility of tumor cells are partly mediated by epithelial-mesenchymal transition (EMT), a process in which cuboidal epithelial cells undergo morphological and biological changes to transition into a mesenchymal phenotype with an elongated and spindle-like shape, thereby allowing the cancer cells to migrate and increasing their metastatic potential [67]. PCa cells (such as LNCaP, C4-2B, PC-3, and DU-145 cell lines) treated with recombinant IGF-1 have an enhanced ability to degrade the underlying basement membrane and invade the ECM owing to EMT [46]. On the one hand, IGF-1 promotes a more mesenchymal phenotype due to the downregulation of E-cadherin levels and inhibition of cell-cell adhesion in PCa (Figure 2) [46,51,68]. This regulation is mediated by forkhead box A1 (FOXA1), a “pioneer factor” in PCa. In response to IGF-1 activation, FOXA1 leaves the nucleus, which prevents its actions on chromatin-packaged DNA. FOXA1 silencing in PCa cells (such as PNT2 and DU145 cell lines) results in a more epithelial-like phenotype manifested by an increase in the epithelial marker E-cadherin and a reduction in the mesenchymal marker vimentin [68,69]. On the other hand, IGF-1 induction enhances the tyrosine phosphorylation of β-catenin and results in the dissociation of β-catenin from the E-cadherin complex because it increases the level of the phosphorylated form of the sarcoma gene (Src). In cancer cells, accumulated β-catenin in the cytoplasm translocates into the nucleus, which has been shown to promote EMT in PCa (Figure 2) [56,68].

#### 3.2.2. Angiogenesis

Angiogenesis means the sprouting of new blood vessels and the formation of a new vascular bed from pre-existing vessels. In tumors, oxygen deprivation and an imbalance of growth factors that facilitate or suppress vessel formation contribute to the formation of an irregular and disorganized vascular network [70]. Signaling through IGF-1R can increase the expression of vascular endothelial growth factor (VEGF) via the activation of hypoxia-inducible factor-1α (HIF-1α) [71]. In mammalian cells, VEGFs are the most well-studied proangiogenic factors, and HIF-1α is a vital transcriptional factor that responds to oxygen deprivation. Under aerobic conditions, HIF-1α is prone to degradation via a ubiquitin-proteasome-dependent pathway. In cancers, however, under hypoxia or stimulation from IGF-1, HIF-1α is stabilized due to the inhibition of prolyl hydroxylase and translocates into the nucleus, where it dimerizes with constitutively expressed HIF-1β, and this HIF-1α/β dimer can bind to hypoxia-response elements of angiogenic growth factors (Figure 2) [71]. The accumulation of HIF-1α protein induced by growth factors enhances VEGF expression, promotes angiogenesis, and achieves increased delivery of oxygen and nutrients to tumors [72].

#### 3.2.3. Bone Metastasis

Malignant tumors can metastasize at any moment and usually target certain organs more than others. Among the various distant organs affected by metastasis in PCa patients, bone metastasis is the leading cause of death and accounts for >80% of metastatic disease cases [73]. PCa bone metastasis mainly involves four steps, namely, cancer cell colonization, dormancy, reactivation and development, and bone reconstruction [74]. This elaborate process is microenvironment-driven, involving intricate interactions between tumor cells and osteoclastic, osteoblastic, and bone stromal cells [74,75]. Prior proliferation of these cells in the bone’s microenvironment is responsible for the preference of bone as a metastasis site [76].

IGF-1 is one of the most abundant growth factors deposited in the bone matrix and can be released during bone resorption and remodeling [77]. Being synthesized by cells such as premature or mature osteoblasts, osteocytes, and osteoclasts, IGF-1 binds to cell surface receptors to stimulate the proliferation and differentiation of osteoblast precursors, thereby enhancing bone formation and maintaining bone mass and creating a bone microenvironment that caters to cancer cell growth (Figure 1) [78,79]. In patients with bone metastasis, IGFs produced by the bone matrix increase the collagenolytic and osteoblast activity of neoplastic cells, thus contributing to metastatic spread (Figure 1) [58,77,80]. This contribution is suspected to be partly associated with lower immune cell infiltration because genes related to immune cells were downregulated after receipt of anti-IGF-1R therapy [58]. The interactions between bone cells and PCa cells can be affected by IGF-1 and play a major role in bone metastasis formation.

### 3.3. Radiosensitivity/Radioresistance

Radiation treatment is a primary treatment option in men diagnosed with localized PCa with no identifiable regional lymph nodes or distant metastasis [81]. This therapy can result in both double- and single-stranded breaks in chromosomal DNA directly or via free radicals [82]. Activation of several tyrosine kinase receptors has been shown to participate in blocking the radiation-induced DNA damage response and mediating radioresistance [83]. Furthermore, suppression of IGF-1R activity has been proven to enhance radiosensitivity in cancer due to delayed double-strand break repair in cancer cells [41].

IGF-1R expression in tumors has been shown to be positively correlated with the probability of recurrence after primary radiotherapy and is associated with adverse outcomes postradiotherapy in PCa [42]. IGF-1R can be rapidly phosphorylated after tumor irradiation, which promotes tumor recovery from radiation [83,84]. In other studies, treatments that target IGF-1R were found to sensitize cancer cells to irradiation, and kinase-deficient mutant IGF-1R cancer cells exhibited significantly decreased growth and viability and reduced clonogenic survival after irradiation compared with wild-type cells [42,85]. The mechanism by which IGF-1R signaling mediates radioresistance is thought to be associated with diminished γ-H2AX foci, which normally promote the resolution of irradiation-induced DNA double-strand breaks (DSBs) via both homologous recombination (HR) and the classical nonhomologous end joining (C-NHEJ) pathway (Figure 2) [41]. IGF-1/IGF-1R activity is recognized as a factor that mediates resistance to radiotherapy in PCa.

### 3.4. Castration Sensitivity/Resistance

Androgen deprivation therapy (ADT) is a part of the first-line therapeutic regimen for advanced or metastatic androgen-dependent PCa; it induces cancer regression, relieves symptoms, and prolongs survival because PCa is an androgen-driven cancer [81,86]. Alterations in AR signaling, such as amplification, expression of splice variants, or posttranslational modifications, caused by ADT are frequent mechanisms that induce cancer resistance to castration [4,86,87,88].

IGF-1/IGF-1R is a candidate autocrine/paracrine ligand-receptor pair that promotes cancer progenitor cell and mature cancer cell resistance to castration [88]. IGF-1/IGF-1R can not only stimulate nuclear translocation of AR and subsequent AR-mediated transcriptional activity, but also affect the AR androgen-binding domain or phosphorylation status to favor the progression of androgen-independent PCa (Figure 2) [26,89,90]. IGF-1 can also enhance androgen signaling even under a very low androgen environment or in the absence of androgen by abolishing the FOXO1 occupancy of AR target gene promoters [91]. Functioning as a tumor suppressor, FOXO1 can be recruited by liganded AR to the chromatin containing its target gene promoters, where it interferes with AR-DNA interactions and disrupts ligand-induced AR subnuclear compartmentalization [91]. Interestingly, ligand-bound AR can also upregulate IGF-1R expression by regulating the Src-Erk1/2 pathway in PCa cells [92]. Ligand-bound AR, which is functionally enhanced by IGF-1 signaling-mediated dissociation of the repressor FOXO1, in turn stimulates the activation of IGF-1R and presumably results in increased tension of IGF-1 signaling, thereby leading to further functional augmentation of the receptor itself [91]. In summary, upregulation or activation of AR and IGF-1R promotes the progression of PCa.

Docetaxel is commonly combined with androgen deprivation in PCa and is usually administered to patients diagnosed with castration-resistant prostate cancer (CRPC). Resistance to docetaxel is a major clinical obstacle in the therapy of advanced PCa. In addition to ADT, increased amounts of IGF-1 secreted from prostatic stromal cells and/or periprostatic stromal tissue also promote the development of resistance to docetaxel in human PCa [5,93]. IGF-1 augments the expression of β-tubulin via enhanced β-tubulin isoform 2B (TUBB2B) expression, and inhibiting miR-143 may be associated with a docetaxel-resistant phenotype [93,94]. Docetaxel exerts anticancer effects by binding to various sites on tubulin and interrupting microtubule dynamics, thus blocking mitosis at the metaphase/anaphase transition. Targeting the IGF-1 axis may overcome ADT or docetaxel resistance and improve the outcome of patients with advanced PCa because both IGF-1 levels and IGF-1R expression are largely unaffected by castration or docetaxel treatment [58].

## 4. IGF-1 Signaling: A Target for PCa Treatment

ADT and second-generation AR signaling inhibitors have been the gold standard of care for advanced or metastatic PCa for decades [27]. Unfortunately, while these treatments initially showed benefits, inevitable resistance to castration and relapse in the form of CRPC, for which there are limited treatment options, are critical events leading to the poor survival of patients with PCa [27].

IGF-1 signaling is frequently dysregulated in cancer development, and its overexpression plays a vital role in the malignant transformation of mammary cells, provides prostate tumors with inherent resistance to radiotherapy or ADT, and worsens the prognosis of patients. Hence, IGF-1/IGF-1R inhibitory reagents have been developed to prevent cancer development and improve survival, including a variety of human neutralizing antibodies, anti-IGF-1R monoclonal antibodies, and a few small molecules. A portion of these agents has been tested in clinical trials alone or in combination with conventional therapies in PCa before radical prostatectomy, PCa before metastasis, and even CRPC [17,95]. Our review mainly covers therapies investigated in PCa patients, and other agents that have potential clinical applicability and have been tested in preclinical experiments will be minimally discussed.

### 4.1. Anti-IGF-1R Monoclonal Antibodies

The initial strategy involves the use of anti-IGF-1R monoclonal antibodies to block ligand-receptor interactions and cause IGF-1R internalization and subsequent degradation [27,96]. Several therapeutic monoclonal antibodies directed against IGF-1R have been developed for use in malignant tumors, such as cixutumumab (IgG1), figitumumab (IgG2), and ganitumab (IgG1). To some extent, the antibodies that have been investigated in clinical trials have shown antitumor growth effects and have been generally well tolerated by most individuals.

#### 4.1.1. Cixutumumab (IMC-A12)

Cixutumumab is a recombinant human IgG1 monoclonal antibody directed at IGF-1R. It was shown to promote the internalization of IGF-1R and induce cell cycle arrest and apoptosis in both androgen-dependent and androgen-independent PCa cells (such as LuCaP 35 and LuCaP 35 V cell lines) [90,97]. There have been several clinical trials combining cixutumumab with other agents for the treatment of patients with various clinical stages of cancer to determine its efficacy and safety in recent decades. However, the results have been somewhat mixed due to the different disease stages and combination drugs used in different trials [90,98,99,100]. For instance, a randomized phase II study combining cixutumumab with ADT showed that it did mildly increase the undetectable PSA (≤0.2 ng/mL) level but did not improve overall survival in newly metastatic hormone-sensitive PCa patients (Table 2) [90,99]. Furthermore, the combination of cixutumumab with mitoxantrone-prednisone resulted in moderate disease control in another phase II trial (Table 2) [100]. Regarding safety, although treatment-related adverse events were reported in patients with PCa, such as low-grade hyperglycemia and fatigue, cixutumumab is generally well tolerated by the majority of patients [90,100]. Mild hyperglycemia may be induced by IGF-1R monoclonal antibodies that bind to IGF-1R/INSR hybrid receptors and downregulate insulin signaling [17]. The efficacy and safety of cixutumumab in treating PCa should be further explored due to the heterogeneity in the findings of these clinical trials.

#### 4.1.2. Figitumumab

Figitumumab is a human IgG2 monoclonal antibody that is highly specific for IGF-1R and blocks the binding of IGF-1. Figitumumab can induce cell cycle arrest, downregulate androgen-regulated gene expression, and increase the sensitivity of tumor cells to chemotherapy [101]. A phase II preoperative study in patients with localized PCa showed that figitumumab decreased both AR expression in PCa and PSA in serum (Table 2) [102]. Another phase II clinical trial in metastatic CRPC showed that patients who received figitumumab plus docetaxel had significantly greater PSA responses than those who received docetaxel alone, but they also had regrettably shorter median progression-free survival (PFS) and more serious treatment-related adverse events than those who received docetaxel alone (Table 2) [101]. These seemingly contradictory findings may be due to the different clinical stages of patients and different treatments of the control groups in different clinical trials. However, the results reveal the challenges of improving the activity of first-line docetaxel monotherapy in CRPC and overcoming cancer resistance to docetaxel. The efficacy and safety of figitumumab in the treatment of PCa should be further studied in additional clinical trials due to the heterogeneity of the results.

#### 4.1.3. A12

A12 is a fully human monoclonal antibody that specifically blocks IGF-1 binding to IGF-1R. It can also downregulate surface IGF-1R expression to induce IGF-1R internalization and degradation in PCa [103]. A study found that A12 causes growth inhibition by increasing apoptosis or G1 cell cycle arrest in androgen-dependent cancers, while increased G2-M cell cycle arrest occurs in androgen-independent cancers due to blocked IGF-1R signaling (LuCaP 35 and LuCaP 35 V cell lines) [97]. In addition, A12 combined with docetaxel can arrest cancer cells in the G2-M cell cycle transition and has been proven to negatively regulate cell cycle progression- and cell survival-associated gene expression, thus decreasing the proliferation of tumor cells [104]. Only some preclinical trials have investigated the efficacy and safety of A12 in individuals with PCa.

#### 4.1.4. Ganitumab (AMG 479)

Ganitumab is a fully human monoclonal antibody that inhibits ligand-induced phosphorylation of IGF-1R and its downstream effectors [17]. Ganitumab treatment resulted in significant downregulation of IGF-1R, thus reducing proliferation and inducing growth suppression in multiple androgen-dependent and androgen-independent PCa xenografts (such as LNCaP and 22Rv1 cells lines) [105,106,107]. Ganitumab reduces IGF-1R activation by binding the L2 domain, a section of the extracellular leucine-rich domain that contributes to ligand binding, thereby preventing both IGF-1 and IGF-2 interaction with the receptor [105]. In addition, this binding also induces internalization and degradation of IGF-1R in vivo [105]. In contrast to other antibodies targeting IGF-1R, ganitumab does not interfere with insulin binding to insulin receptor (INSR) homodimers and instead inhibits the formation of hybrid IGF-1R/INSR receptors [108]. In several phase I trials, AMG 479 was well tolerated and could be safely administered at the highest planned dose of 20 mg/kg [109]. Toxicologic evaluation of AMG 479 in advanced malignancies revealed primary acceptable toxicity events, mainly including mild anemia, transient thrombocytopenia, and fatigue [109,110]. Nevertheless, we could not find any clinical trials of ganitumab as a treatment for patients suffering from PCa or any efficacy assessments.

#### 4.1.5. Other IGF-1R-Targeting Monoclonal Antibodies

There are other IGF-1R-targeting monoclonal antibodies that have been applied in cancer therapy. For instance, BIIB022 is a nonglycosylated human anti-IGF-1R monoclonal antibody of the IgG4 isotype and exerts its inhibitory effect via an allosteric mechanism instead of competing with the ligand [111]. A phase I trial illustrated that responses to BIIB022 were heterogeneous: 55% of patients achieved stable disease, 29% had progressive disease, and 12% were not evaluable for response [111]. In addition, dalotuzumab is a humanized IgG1 monoclonal antibody that binds to IGF-1R, induces receptor internalization and degradation, and inhibits IGF-mediated cell proliferation [112]. A clinical trial in solid tumors illustrated that patients had a mixed response that ranged in duration from 13 to 43 weeks [112]. In addition, robatumumab is a fully human anti-IGF-1R monoclonal antibody of the IgG1 isotype. It binds to the extracellular portion of human IGF-1R with high affinity, prevents IGF binding and activation of transduction events, and has been shown to inhibit tumor growth in various human tumor xenograft models via inhibition of cancer cell proliferation and blood vessel formation [113]. These monoclonal antibodies have been generally well tolerated in patients with solid tumors. Treatment-related adverse events, such as hyperglycemia, fatigue, and abdominal pain, were mild to moderate in severity (grades ≤ 2) and mostly reversible [111,112,114,115]. However, we did not find any clinical trials or even preclinical experiments assessing these antibodies in PCa.

**Table 2 cancers-15-01287-t002:** Literature of IGF-1R inhibitors under clinical investigation in PCa.

Agents	Property	Phase/Region	Population	Treatment Schemes	Key Activity Data	Safety and Adverse Events	References
Cixutumumab	Recombinant human mAb, IgG1, targets IGF-1R	Phase II (2015/2020/USA)	210 patients with mHSPC (PSA ≥ 5 ng/mL)	Cixutumumab (10 mg/kg, intravenously, 2 times/28 d, last 28 w) + Bicalutamide (oral, daily) + LHRH agonist	Increase the undetectable PSA (≤0.2 ng/mL) ratio, but not improve overall survival	Well tolerated; Minimal increase in adverse events such as hyperglycemia.	[90,99] (J Clin Oncol, 2015); (Prostate Cancer and Prostatic Diseases, 2020)
		Phase I (2019/USA)	16 patients with mCRPC (PSA ≥ 2 ng/mL)	Cixutumumab (6 mg/kg, 12 w) + Temsirolimus	Limited antitumor effect with no PSA decreases ≥ 50%.	Hyperglycemia (grade ≤ 3); No grade 4 events or deaths occurred.	[98] (Clinical Genitourinary Cancer, 2020)
		Phase II (2015/USA)	132 patients with docetaxel-treated mCRPC (PSA ≥ 2 ng/mL)	Cixutumumab (6 mg/kg, intravenously,3 times/21 d) + Mitoxantrone + Prednisone	A PSA decline of ≥ 50% from baseline occurred in 18.5% of patients; The median cPFS was 4.1 months and median OS was 10.8 months.	Safety in the mass; Fatigue and weight decreased mildly.	[100] (Eur J Cancer, 2015)
Figitumumab	Full human mAb, IgG2, blocks IGF-1 binds to IGF-1R	Phase II (2012/Canada)	16 patients with localized PCa (PSA ≥ 10 ng/mL)	Figitumumab (20 mg/kg, intravenously every 3 weeks, 3 cycles)	Decline serum PSA and testosterone levels; inhibit androgen receptor expression.	Mild adverse events (grade ≤ 3).	[102] (Clin Cancer Res, 2012)
		Phase II (2014/UK)	204 patients with mCRPC	Figitumumab (10 or 20 mg/kg, intravenously, 12 m) + Docetaxel vs. Docetaxel alone	Higher stable PSA response rate (31% vs. 21%) Shorter median PFS (4.9 months vs. 7.9 months).	More treatment-related serious adverse events (41% vs. 15%); Hyperglycemia, diarrhea, asthenia.	[101] (Clin Cancer Res, 2014)

Abbreviations: mAb, monoclonal antibody; OS, overall survival; PFS, progression-free survival; d, days; w, weeks; m, months; PSA, prostate-specific antigen; mHSPC, metastatic castration-sensitive prostate cancer; CRPC, castration-resistant prostate cancer.

### 4.2. IGF-Neutralizing Antibodies

IGF-neutralizing monoclonal antibodies target IGF ligands to neutralize proliferative and prosurvival signaling triggered by both ligands. IGF-1/-2-neutralizing monoclonal antibodies inhibit various IGF signaling pathways at the same time but have a lower risk of hyperglycemia than IGF-1R/INSR tyrosine kinase inhibitors, as they do not affect insulin receptor-β, which regulates glucose homeostasis [116]. Several therapeutic monoclonal antibodies directed against IGF-1 have been developed for the treatment of malignant tumors, such as xentuzumab (IgG1) and dusigitumab (IgG2). These antibodies have shown antitumor growth activities when investigated in preclinical experiments. However, there have been no clinical trials to determine their efficacy and safety in patients with PCa.

#### 4.2.1. Xentuzumab (BI 836845)

Xentuzumab is a fully humanized IgG1 monoclonal antibody that targets IGF-1 and IGF-2 with high affinity and potently blocks the biological effects mentioned above [116,117]. Xentuzumab inhibits the phosphorylation of well-established direct substrates of FOXO transcription factors, which are important regulators of the expression of genes involved in survival [118]. Binding of xentuzumab to IGFs leads to a lower degradation rate for bound than for free IGFs. In turn, the reduction in free IGF-1 triggers compensatory mechanisms, which increase IGF-1 synthesis to restore homeostasis, thereby leading to increases in total IGF-1. However, the free IGF-1 levels are reduced [117,119]. In human phase I trials, xentuzumab was well tolerated at all investigated doses [116,119]. In advanced/metastatic breast cancer, a PFS benefit was observed in patients without visceral metastasis treated with xentuzumab/everolimus/exemestane compared with patients treated without xentuzumab [120]. In addition, enzalutamide is a low-mass oral androgen receptor inhibitor that targets multiple steps in the androgen receptor signaling pathway and has shown significant benefits to patients with advanced prostate cancer [121]. A study reported that xentuzumab combined with enzalutamide could be effective in patients who are resistant to enzalutamide alone and may be able to be used to overcome castration resistance [118].

#### 4.2.2. Dusigitumab (MEDI-573)

Dusigitumab is a fully human IgG2 monoclonal antibody that inhibits IGF signaling through both the IGF-1R and INSR-α pathways by neutralizing IGF ligands. Dusigitumab was isolated from mice that were alternately immunized with soluble recombinant human IGFs and lacked cross-reactivity to human insulin [122]. A clinical trial that enrolled 43 individuals with advanced solid tumors illustrated that dusigitumab has antitumor ability and has less impact on metabolic homeostasis than anti-IGF-1R therapies [123]. Dusigitumab can inhibit the phosphorylation of IGF-1R without changing the levels of total proteins. A tumor xenograft model suggested that the combination of dusigitumab and antibodies targeting the IGF receptor may block angiogenesis in IGF-driven tumors but not inhibit tumor growth due to its low binding affinity for circulating IGF-1 [124].

### 4.3. IGF-1R Inhibitory Small-Molecule Agents

IGF-1R inhibitory small-molecule agents are usually ATP-competitive antagonists. They have been described as dual IGF-1R/INSR inhibitors because the ATP-binding site is homologous between the IGF-1R and INSR kinase domains [125]. Due to potential issues related to this homology, the development of IGF-1R inhibitors as anticancer therapeutics has focused mainly on the identification of antibodies specific for IGF-1R as mentioned above. We will describe some IGF-1R inhibitory small-molecule agents, such as linsitinib and BMS-754807 on their antitumor effects.

#### 4.3.1. Linsitinib (OSI-906)

Linsitinib is an oral small-molecule tyrosine kinase inhibitor of IGF-1R and INSR that has shown preliminary evidence of antitumor activity [126]. Linsitinib can inhibit tumor growth in an IGF-1R-driven xenograft mouse model [127]. However, it has not performed well in several solid tumors, as shown in several clinical trials [126,128,129,130]. A phase II study to determine the clinical role of linsitinib in men with metastatic CRPC showed that linsitinib had an acceptable tolerability profile, with the majority of adverse events being mild nausea, fatigue, and diarrhea. However, this study was unable to confirm the preclinical results supporting the inhibition of IGF-1R in this setting because no significant PSA or objective response was shown following two cycles of treatment [128]. Clinically, multidrug resistance (MDR) limits the efficacy of anticancer drugs and the possibility of successful cancer chemotherapy [131]. Intriguingly, linsitinib can reverse MDR mediated by ATP-binding cassette transporters, which can reduce the intracellular accumulation of chemotherapeutic drugs by increasing the efflux of drugs from cancer cells [132]. This theory implies that linsitinib might be beneficial when added to traditional chemotherapy because it can overcome MDR, but this hypothesis needs to be further verified by multiple animal experiments.

#### 4.3.2. BMS-754807

BMS-754807 is a pyrrolotriazine and reversible ATP-competitive antagonist of IGF-1R that inhibits the catalytic function of IGF-1R [125,133]. Given the high homology of the kinase domains (approximately 85%) of IGF-1R and INSR, dysregulation of glucose homeostasis is a challenge [125]. BMS-754807 inhibits cancer cell proliferation and induces cell apoptosis by inhibiting Akt phosphorylation in an orthotopic PC3-LG cell mouse model and suppresses tumor growth [134]. These strong antiproliferative and proapoptotic effects are mediated by induction of G2/M cycle arrest and activation of caspase-3 and PARP in cancers prone to poor prognosis [125]. There have been no clinical trials to determine the efficacy and safety of BMS-754807 in PCa patients yet.

#### 4.3.3. Picropodophyllin (AXL1717/PPP)

Picropodophyllin (PPP) is a known inhibitor of IGF-1R tyrosine phosphorylation and has been proven effective in inhibiting tumor growth in vivo [135]. In PCa, PPP can induce cell cycle arrest and apoptosis via the production of ROS and inhibition of the PI3K/AKT signaling pathway, thereby inhibiting cancer development [136]. At present, there are only phase I and II clinical trials of PPP in non-small-cell lung cancer (NSCLC) patients.

### 4.4. Brief Summary

Abundant preclinical experiments have been performed in both cancer cells and tumor xenograft models, and they have confirmed that drugs and antibodies targeting IGF-1/IGF-1R signaling have antitumor growth effects. Most inhibitory reagents, including monoclonal antibodies and tyrosine kinase inhibitors, have not yet displayed significant benefits in patients in phase II clinical trials. This limited antitumor effect is probably the result of the scarcity of reliable predictive biomarkers that predict response to IGF-1R inhibition [17,137]. For example, neuropilins (NRPs) are one class of VEGF receptors that are particularly valuable with respect to angiogenesis and cancer biology. Studies have found that the expression of NRP2 in PCa cells contributes to tumor formation, facilitates cancer metastasis, and is correlated with Gleason grade [138,139]. Surprisingly, because prostate carcinomas that express NRP2 exhibit low levels of IGF-1R, NRP2 expression in tumor cells may be a potential biomarker for predicting the efficacy of IGF-1R inhibitors [138]. Therefore, IGF-1R is still a valid investigational target for patients with PCa. Combinations of IGF-1R inhibitors with other targeted agents may improve treatment efficacy, result in partial responses, or reduce tumor volume. Furthermore, some newer agents for metastatic hormone-sensitive PCa may require more extensive preclinical testing in multiple models and need to demonstrate better synergistic effects with standard-of-care agents before introduction into clinical trials.

Future therapeutic strategies might include molecules that are capable of interfering with IGFBPs, small-molecule competitive binding antagonists, monoclonal antibodies against IGF-1, antisense oligonucleotides (ASOs), siRNAs, dominant-negative IGF-1R-specific tyrosine kinase inhibitors, and inhibitors of IGF-1R downstream effectors [24].

## 5. Conclusions

This review summarizes the contribution of IGF-1/IGF-1R signaling to the development of PCa and highlights the relevance of IGF-1/IGF-1R signaling in potential cancer therapies. In summary, there is substantial evidence regarding the critical role of IGF-1/IGF-1R signaling in the progression of PCa, with studies reporting correlations of IGF-1/IGF-1R with tumor cell proliferation, Gleason score, metastatic disease progression, and clinical outcomes. Targeting IGF-1R has been extensively investigated as a strategy in cancer therapy. Several studies have also demonstrated the efficacy and safety of anti-IGF-1R agents in preclinical PCa models or clinical trials. However, therapies that target IGF-1R have shown relatively limited benefits in clinical trials. Based on previous research, we hypothesize that the limited IGF-1 efficacy in prostate cancer in humans is probably due to the scarcity of reliable predictive biomarkers that forecast response to IGF-1R inhibition. Research to better understand detailed mechanisms of IGF-1/IGF-1R signal in the development of PCa and explore biomarkers to predict response and prognosis are warranted for personalized treatments and follow-up strategies. Concurrently, IGF-1R targeting in combination with other existing therapeutic options, including cytotoxic agents, radiotherapy, and castration, ought to be attempted to look for a novel strategy for the treatment of prostate cancer. Furthermore, developing modified reagents that target IGF-1/IGF-1R may offer a novel therapeutic regimen.

## Figures and Tables

**Figure 1 cancers-15-01287-f001:**
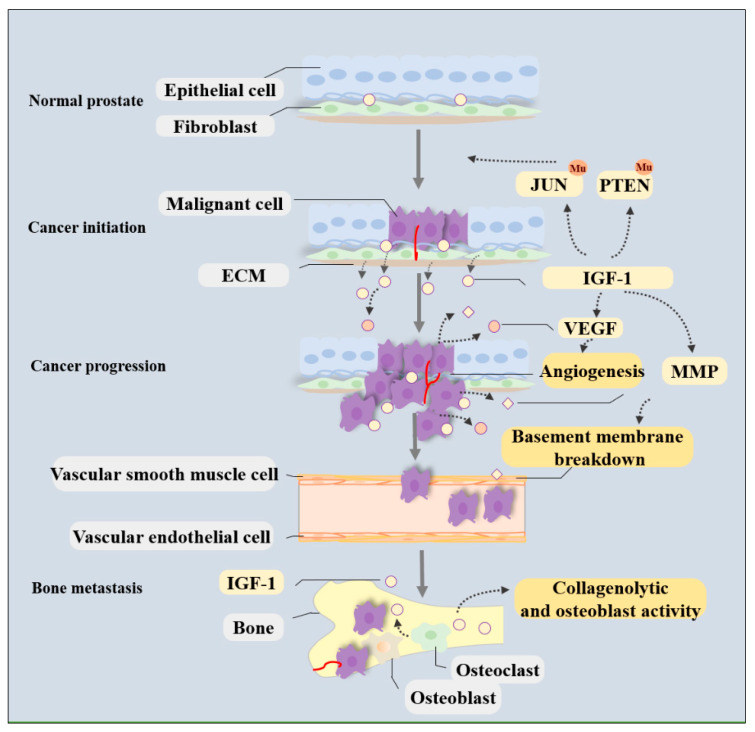
Schematic representation of the IGF-1 signaling on promoting PCa development in tumor niche. Autocrine and paracrine IGF-1 signaling from the malignant cells, fibroblast, and ECM can contribute to tumor growth and distant metastasis in PCa. IGF-1 can induce by regulating the mutation of oncogenes and tumor suppressor genes. It can induce angiogenesis in the tumor microenvironment by enhancing VEGF expression and promoting degradation of ECM components by stimulating MMP release to favor metastasis of cancer cells. In addition, IGF-1 produced by the bone matrix can increase the collagenolytic and osteoblast activity of neoplastic cells, thus contributing to bone metastasis formation.

**Figure 2 cancers-15-01287-f002:**
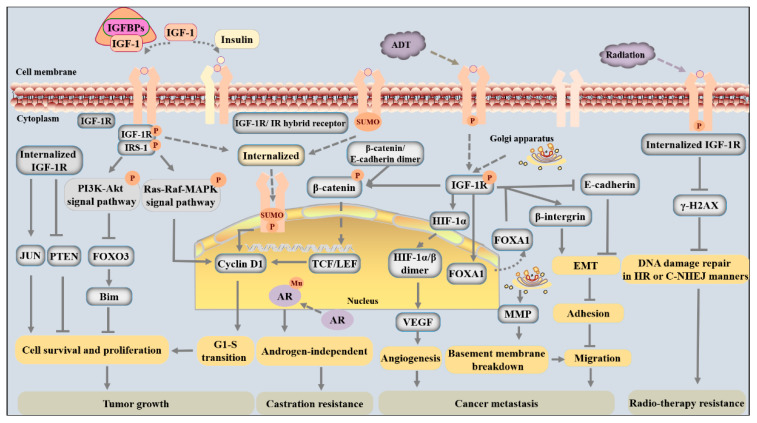
The detailed molecular mechanisms of activated IGF-1R in promoting cancer development and therapy resistance are schematically represented. IGF-1 exerts its biological functions via activating PI3K-Akt as well as Ras-MAPK signaling pathways via promoting phosphorylation and SUMOylation of its receptor IGF-1R. First of all, activated IGF-1R can promote survival and proliferation and inhibit apoptosis of cancer cells by regulating the expression of cell cycle proteins, oncogenes, and tumor suppressor genes, such as *cyclin D1*, *JUN*, and *PTEN*. Secondly, activated IGF-1R can induce angiogenesis in the tumor microenvironment by enhancing VEGF expression. It can also promote cancer cell migration via regulating cell adhesion-associated proteins, and then promoting EMT. In addition, IGF-1R can promote degradation of ECM components by stimulating MMP release. All these biological functions would enhance the ability of metastasis for cancer cells. Thirdly, IGF-1/IGF-1R would favor the progression of androgen-independent PCa due to stimulating nuclear translocation of AR and affecting the AR androgen-binding domain or phosphorylation status for mutation. Lastly, activated IGF-1R can promote DNA damage repair and radioresistance of cancer cells by inhibiting DNA damage-related proteins and promoting castration resistance for androgen receptor mutation.

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
