# Peer review of "Emerging Role of IGF-1 in Prostate Cancer: A Promising Biomarker and Therapeutic Target"

_cancers, 2023, doi:10.3390/cancers15041287_

Round 1

Reviewer 1 Report

The review manuscript "Emerging Role of IGF-1 in Prostate Cancer: A Promising Biomarker and Therapeutic Target" is a comprehensive review in the field. The study presented here is of broad scientific interest group.

However,

1. Most of the studies that are discussed here in the introduction is very old and does not reflect the recent advancement in the field.

2. Table1 & Table 2 are unorganized and difficult to read.

3. Figure. 1 is very busy, most of the things are irrelevant and difficult for the readers to understand.

Conclusion needs to be rewritten and should also reflect upon the future avenues in the field.

Author Response

Response to the comments     

Dear reviewer,

    We would like to thank you for your careful reading and helpful comments about the review“Emerging Role of IGF-1 in prostate cancer: A Promising Biomarker and Therapeutic Target”(Manuscript ID: cancers-2126944). We have carefully considered all comments from you and revised our manuscript accordingly. All the errors you picked and recommendations you proposed are greatly helpful for us to polish our manuscript. Our manuscript has been checked and polished by AJE company, and it has been submitted to editor.

    In the follow sections, we summarize our responses point by point to each comment from you. We hope that our responses have well addressed all concerns from you and our revised manuscript can be accepted for publication.

Question 1: Most of the studies that are discussed here in the introduction is very old and does not reflect the recent advancement in the field.

Response 1: In the section 1.1-1.4, we mainly summarize epidemiological studies about the relationship between circulating IGF-1 levels and local IGF-1R expression to PCa development and outcome comprehensively. Multiple early-stage epidemiological researches have elaborated that serum IGF-1 level and the incidence rate of prostate cancer. For example, “Plasma insulin-like growth factor-I and prostate cancer risk: a prospective study” argued that“plasma IGF-I as a predictor of prostate cancer risk may have implications for risk reduction and treatment”(Reference 20, line 87 - line 91, Science 1998; 279(5350):563-566). However, the latest studies had focused on the relationship between IGF-1R and the outcome of PCa. For example, an article named “Insulin signaling, androgen receptor and PSMA immunohistochemical analysis by semi-automated tissue microarray in prostate cancer with diabetes” held that “IGF-1 receptor was associated with Prostate-Specific Membrane Antigen, Ki-67 and IR-β” (Reference 29, line 92 - line 94, Translational Research, 2021; 238:25-35). Another article named “Expression of IGF/insulin receptor in prostate cancer tissue and progression to lethal disease” found that“Tumors with strong IGF-1R expression showed increased cell proliferation, decreased apoptosis and a higher prevalence of ERG. Strong IGF-1R was associated with a borderline increased risk of lethal prostate cancer”(Reference 28, line 92 - line 94, Carcinogenesis 2018; 39(12):1431-1437). Epidemiological studies have gradually transitioned from the relationship between serum IGF-1 and the incidence of prostate cancer to the local level of IGF-1 or the activation of IGF-1R and the outcome or of prostate cancer.

In the section 3.1-3.4, we mainly summarize some preclinical researches or clinical trails about IGF-1/IGF-1R inhibitory reagents on patients with prostate cancer. For example, in some studies, “Cixutumumab” and “figitumumab”shown limited anti-tumor effects in clinical trails (Reference 101Phase II randomized study of figitumumab plus docetaxel and docetaxel alone with crossover for metastatic castration-resistant prostate cancer. Clin Cancer Res 2014; 20(7):1925-1934; Reference 99Survival outcomes and risk group validation from SWOG S0925: a randomized phase II study of cixutumumab in new metastatic hormone-sensitive prostate cancer. Prostate cancer and prostatic diseases 2020; 23(3):486-493Table 2). However, some novel antibody or inhibitory small-molecule agents targets to IGF-1R had also been developed and associated researches were published in the short-term. For example, “xentuzumab” is a novel IGF-neutralizing antibody and it had been applied in multiple clinical trails on patients with solid tumors. Although these reagents have not been applied in prostate cancer in clinical, it has shown antitumor effects in breast cancer or other solid tumors and pre-clinical prostate cancer. (Reference119, A phase 1 trial of xentuzumab, an IGF-neutralizing antibody, in Japanese patients with advanced solid tumors. Cancer Sci 2022; 113(3):1010-1017; Reference118, Antitumor Activity of the IGF-1/IGF-2-Neutralizing Antibody Xentuzumab (BI 836845) in Combination with Enzalutamide in Prostate Cancer Models. Molecular cancer therapeutics 2020; 19(4):1059-1069 ).

We began with the early researches to introduce the relationship between IGF-1/IGF-1R signal and the incidence or outcome of prostate cancer and aim to introduce its potential value as a therapeutic target on prostate cancer more systematically.

Question 2: Table1 & Table 2 are unorganized and difficult to read.

Response 2: Thanks very much for your earnest advice and we are very sorry for the uncoordinated tables. In our updated manuscript, the bullets in the tables have been replaced with Arabic numerals, and some contents have been simplified slightly. In addition, we have adjusted the vertical tables to horizontal tables for the convenience of readers (Table 1: page 4-5 of 26, Table 2: page 4-5 of 26). The details are shown in our revised manuscript.

Question 3: Figure. 1 is very busy, most of the things are irrelevant and difficult for the readers to understand.

Response 3: We are sorry for our negligence of poor resolution and crowded setting in the figure. We have split out panels A and B into two separate figures (figures 1 and 2) and improved the resolution of these figures in the revised manuscript (page10-11 of 26). In addition, we have optimized the content and replenished figure legends below the corresponding figures to make it easier for readers to understand the meaning of the diagram. We have also tagged some cross-references within the text to better join-up the detail in the Figures with various parts of the text as your recommendation. (Figure 1 cross-references in line 166, line 217, line 242, line 293, line 295 and Figure 2 cross-references in line 167, line 187, line 195, line 208, line 251, line 260, line 274, line 318, line 332). The details are shown in our revised manuscript.

Question 4: Conclusion needs to be rewritten and should also reflect upon the future avenues in the field.

Response 4: Thanks for your advice on our conclusion. We have re-written the latter part of concluding remarks according to the suggestions from you. The updated latter part of concluding remarks is as follows. “Based on previous researches, we hypothesize that the limited IGF-1 effiency on prostate cancer in humans probably due to the scarcity of reliable predictive biomarkers that forecast response to IGF-1R inhibition. Research to better understand detailed mechanisms of IGF-1/IGF-1R signal in the development of PCa and explore biomarkers to predict response and prognosis are warranted for personalized treatments and follow-up strategies. Concurrently, IGF-1R targeting in combination with other existing therapeutic options, including cytotoxic agents, radiotherapy and castration, ought to be attempted to look for a novel strategy for the treatment of prostate cancer. Furthermore, developing modified reagents that target to IGF-1/IGF-1R may offer a novel therapeutic regimen”. While the previous contents in this part is as follows. “Biomarkers to predict prognosis and stratifying patients by risk of progression are important for personalized treatments and follow-up strategies. New endeavors should focus on the creation of dual oral tyro-sine kinase inhibitors that target IGF-1R for the treatment of metastatic CRPC due to the lower objective response rates seen with other monotherapies in this disease”. We hope that our concluding remarks can provide a blueprint for seeking a novel strategy for the treatment of prostate cancer.

Thanks again for your reading our manuscript carefully and giving the above helpful comments.

Sincerely,

Feng Pan, M.D./Ph.D.
Department of Urology
Union Hospital, Tongji Medical College
Huazhong University of Science & Technology
1277 Jiefang Avenue
Wuhan  430022
China

Reviewer 2 Report

This review has good merit. The authors have forensically mined the (substantial) Literature regarding the role of insulin growth factor (IGF-1) and its associated receptor (IGF-1R) and have provided the evidence base, purposefully concentrating mainly on patient studies, to underscore their importance as potential biomarker(s) and to provide an assessment of the IGF-1 as a valid target for continued therapeutic intervention. The subject matter will be of interest to specialists in the context of prostate cancer therapeutics and for those investigators newly entering the area. In the latter regard, and to capture the interest of scientists whose interests reside in the broader development of biomarkers of disease, I recommend (whilst acknowledging abbreviations are defined within the text, and some below the Table) that a separate list abbreviations at end of the manuscript would be beneficial to those less familiar.

Overall, the information is logically presented, relevant and easy to follow. Figure 1 has a lot of (valid) detail in each of the panels A and B but currently only one cross-reference to the narrative (line 165); a recommendation is to consider splitting out panels A and B into two separate Figures – which would provide greater clarity for the reader, and also permit further cross-references to other statements that are present in the narrative throughout Section 1, plus sections 2.2.1 and 2.2.2. This would better join-up the detail in the Figure(s) with various parts of the text.

The sections (3. – 3.2.2) summarizing the early work and current status of antibody approaches to target IGF-1/IGF-1R, both clinical and pre-clinical studies, are well-written. 

Section 3.2.1 Xentuxumab

This section describes the IGF-1/IGF-2 targeting antibody xentuxumab but also introduces a combination study (Ref 118. Weyer-Czernilofsky et al., 2020) with enzalutamide – which is a low molecular mass drug. Given subsequent sections lead on to low mass agents, it would better to clarify (rather than assume knowledge) that enzalutamide is a low mass oral Androgen Receptor inhibitor that inhibits multiple steps in the AR signaling pathway (as shown by Tran et al 2009; and referenced by Saad et al 2013).

Tran, C., Ouk, S., Clegg, N., Chen, Y., Watson, P., Arora, V. et al. (2009) Development of a second generation antiandrogen for treatment of advanced prostate cancer. Science 324: 787–790.

Saad F. Evidence for the efficacy of enzalutamide in post chemotherapy metastatic castrate-resistant prostate cancer. Ther Adv Urol. (2013) 201-10. doi: 10.1177/1756287213490054

Recommendation: Section 3.3 IGF-1R inhibitory reagents – would be more accurately described as IGF-1R inhibitory small-molecule agents.

Section 3.3.2 BMS-754807

This section describes another low mass agent and makes the valid point that as a non-specific IGF-1R and IR inhibitor, BMS-754807 runs the risk of adverse effects on glucose homeostasis. However, this would also apply to the low mass drug Linsitinib, described in the preceding section 3.3.1.

Consider the repositioning of this point to 3.2.1 and making the point that it equally applies to BMS-754807. Perhaps consider/clarify the use of IR versus INSR in different parts of the manuscript, if they both refer to the insulin receptor?

Concluding remarks

The final sentence under concluding remarks should be clarified with regard to what is meant by “dual” oral tyrosine kinase inhibitors. Do the authors mean TK inhibitors that target both IGF-1 and IGF-1R, or something else? This is important because of the caveats that are noted with regard to a lack of specificity. If the authors stress this as a focus, the then I am not sure the case has been made evident, and as it stands, conflicts somewhat with lines 558 – 561 outlining future therapeutic strategies, and to some extent the Abstract which points towards a dual approach of targeting IGF-1R with either androgen deprivation or cytotoxics to provide better patient outcomes. I advise a reappraisal of the ‘take-home messages’ the authors wish to convey, in what is an otherwise very commendable review.

Author Response

Response to the comments

Dear reviewer,

    We would like to thank you for your careful reading, helpful comments and constructive suggestions, which has a significant improved the presentation of our manuscript“Emerging Role of IGF-1 in prostate cancer: A Promising Biomarker and Therapeutic Target”(Manuscript ID: cancers-2126944). We have carefully considered all comments from you and revised our manuscript accordingly. In the follow sections, we summarize our responses point by point to each comment from you. We hope that our responses have well addressed all concerns from you and our revised manuscript can be accepted for publication.

Question & Recommendation 1: In the latter regard, and to capture the interest of scientists whose interests reside in the broader development of biomarkers of disease, I recommend (whilst acknowledging abbreviations are defined within the text, and some below the Table) that a separate list abbreviations at end of the manuscript would be beneficial to those less familiar.

Response 1: We gratefully appreciate for your valuable suggestion and have provided a list of abbreviations and arranged alphabetically at end of the manuscript (line 627-line 644).

Question & Recommendation 2: Overall, the information is logically presented, relevant and easy to follow. Figure 1 has a lot of (valid) detail in each of the panels A and B but currently only one cross-reference to the narrative (line 165); a recommendation is to consider splitting out panels A and B into two separate Figures – which would provide greater clarity for the reader, and also permit further cross-references to other statements that are present in the narrative throughout Section 1, plus sections 2.2.1 and 2.2.2. This would better join-up the detail in the Figure(s) with various parts of the text.

Response 2: We are sorry for our negligence of poor resolution of the figure and thank you for earnest recommendation about this problem. We have split out panels A and B into two separate figures (figures 1 and 2) in the revised manuscript (page10-11 of 26). In addition, we have tagged some cross-references within the text to better join-up the detail in the Figures with various parts of the text as your recommendation. (Figure 1 cross-references in line 166, line 217, line 242, line 293, line 295 and Figure 2 cross-references in line 167, line 187, line 195, line 208, line 251, line 260, line 274, line 318, line 332). 

Question & Recommendation 3:

The sections (3–3.2.2) summarizing the early work and current status of antibody approaches to target IGF-1/IGF-1R, both clinical and pre-clinical studies, are well-written.

Section 3.2.1 Xentuxumab. This section describes the IGF-1/IGF-2 targeting antibody xentuxumab but also introduces a combination study (Ref 118. Weyer-Czernilofsky et al., 2020) with enzalutamide – which is a low molecular mass drug. Given subsequent sections lead on to low mass agents, it would better to clarify (rather than assume knowledge) that enzalutamide is a low mass oral Androgen Receptor inhibitor that inhibits multiple steps in the AR signaling pathway (as shown by Tran et al 2009; and referenced by Saad et al 2013).

Tran, C., Ouk, S., Clegg, N., Chen, Y., Watson, P., Arora, V. et al. (2009) Development of a second generation antiandrogen for treatment of advanced prostate cancer. Science 324: 787–790.

Saad F. Evidence for the efficacy of enzalutamide in post chemotherapy metastatic castrate-resistant prostate cancer. Ther Adv Urol. (2013) 201-10. doi: 10.1177/1756287213490054

Response 3: Thank you very much for pointing out this issue. We have described that “enzalutamide is a low mass oral androgen receptor inhibitor that has shown significant benefits to patients with advanced prostate cancer”,as the reference“Saad F. Evidence for the efficacy of enzalutamide in post chemotherapy metastatic castrate-resistant prostate cancer. Ther Adv Urol. (2013) 201-10. doi: 10.1177/1756287213490054”have introduced (section3.2.1, line526-line528).

Question & Recommendation 4:

Recommendation: Section 3.3 IGF-1R inhibitory reagents – would be more accurately described as IGF-1R inhibitory small-molecule agents.

Response 4: We have corrected “IGF-1R inhibitory reagents” as “IGF-1R inhibitory small-molecule agents” according to your comment (line 542). IGF-1R inhibitory reagents including human neutralizing antibodies, anti-IGF-1R monoclonal antibodies and a few small molecules (represented in line 398). We are sorry for inaccurately used this concept in the preliminary manuscript.

Question & Recommendation 5:

Section 3.3.2 BMS-754807

This section describes another low mass agent and makes the valid point that as a non-specific IGF-1R and IR inhibitor, BMS-754807 runs the risk of adverse effects on glucose homeostasis. However, this would also apply to the low mass drug Linsitinib, described in the preceding section 3.3.1. Consider the repositioning of this point to 3.2.1 and making the point that it equally applies to BMS-754807. Perhaps consider/clarify the use of IR versus INSR in different parts of the manuscript, if they both refer to the insulin receptor?

Response 5: Thank you for pointing out this problem. Both IR and INSR mentioned in the manuscript are refer to insulin receptor. We have consistent modified “IR” to “INSR” in the revised manuscript (Table 1, line 533, line 570). 

Question & Recommendation 6:

Concluding remarks

The final sentence under concluding remarks should be clarified with regard to what is meant by “dual” oral tyrosine kinase inhibitors. Do the authors mean TK inhibitors that target both IGF-1 and IGF-1R, or something else? This is important because of the caveats that are noted with regard to a lack of specificity. If the authors stress this as a focus, the then I am not sure the case has been made evident, and as it stands, conflicts somewhat with lines 558 – 561 outlining future therapeutic strategies, and to some extent the Abstract which points towards a dual approach of targeting IGF-1R with either androgen deprivation or cytotoxics to provide better patient outcomes. I advise a reappraisal of the ‘take-home messages’ the authors wish to convey, in what is an otherwise very commendable review.

Response 6:

We are very sorry for the misunderstanding caused by inappropriate use of this word. “Dual”was interpreted as both “IGF-1R”and“INSR”tyrosine kinase inhibitors originally.

However, for consistant with abstract and outlining future therapeutic strategies (lines 558 – 561 ), we re-written the latter part of concluding remarks according to the suggestions from you. The latter part of concluding remarks is as follows. “Based on previous researches, we hypothesize that the limited IGF-1 effiency on prostate cancer in humans probably due to the scarcity of reliable predictive biomarkers that forecast response to IGF-1R inhibition. Research to better understand detailed mechanisms of IGF-1/IGF-1R signal in the development of PCa and explore biomarkers to predict response and prognosis are warranted for personalized treatments and follow-up strategies. Concurrently, IGF-1R targeting in combination with other existing therapeutic options, including cytotoxic agents, radiotherapy and castration, ought to be attempted to look for a novel strategy for the treatment of prostate cancer. Furthermore, developing modified reagents that target to IGF-1/IGF-1R may offer a novel therapeutic regimen”. We aim to describe some concluding remarks that may provide a blueprint for seeking a novel strategy for the treatment of prostate cancer.

Thanks again for your reading our manuscript carefully and giving the above positive comments.

Sincerely,

Feng Pan, M.D./Ph.D.
Department of Urology
Union Hospital, Tongji Medical College
Huazhong University of Science & Technology
1277 Jiefang Avenue
Wuhan  430022
China

Reviewer 3 Report

Article title: Emerging Role of IGF-1 in prostate cancer: A Promising Biomarker and Therapeutic Target

The authors in this review have summarized the role of IGF-1/IGF-1R signaling in prostate cancer development and progression. They have also described in detail the mechanism of IGF-1 signaling in different stages of prostate cancer growth including cell migration, cell adhesion, angiogenesis, and metastasis. This review also provides information about the current therapeutic strategies for targeting IGF-1 signaling and the ongoing clinical trials.

The following minor points need to be considered:

1. The alternate name of the drug Cixutumumab (IMC-A12) can be mentioned since the authors have mentioned an alternate name for almost all the drugs used in the clinical trial.

2. Kindly cite Table 1 and Table 2 in the respective section of the manuscript.

3. The resolution of the figure needs to be improved.

4. It is better to mention cell lines used for checking the effect of the drug. For example, in the case of Cixutumunab (Line number 382-384), the author mentions “It was shown to promote the internalization of IGF-1R and induce cell cycle arrest and 383 apoptosis in both androgen-dependent and androgen-independent PCa cells”. It is better if the cell lines used in the study are mentioned wherever necessary.

Author Response

Response to the comments

Dear reviewer,

    We would like to thank you for your careful reading, helpful comments and constructive suggestions, which has a significant improved the presentation of our manuscript “Emerging Role of IGF-1 in prostate cancer: A Promising Biomarker and Therapeutic Target”(Manuscript ID: cancers-2126944). We have carefully considered all comments from you and revised our manuscript accordingly. In the follow sections, we summarize our responses point by point to each comment from you. We hope that our responses have well addressed all concerns from you and our revised manuscript can be accepted for publication.

Question & Recommendation 1: The alternate name of the drug Cixutumumab (IMC-A12) can be mentioned since the authors have mentioned an alternate name for almost all the drugs used in the clinical trial.

Response 1: We gratefully appreciate for your detailed suggestion and have added” IMC-A12” to corresponding section (Section 3.1.1, line 412).

Question & Recommendation 2:  Kindly cite Table 1 and Table 2 in the respective section of the manuscript.

Response 2:  Thanks for your earnest advice. In our updated manuscript, we have cite these tables in proper position. Table 1 cross-references in Section 1, line 82 and Table 2 cross-references in Section3.1.1 - 3.1.2, line 422 – 424, line 436- 441).

Question & Recommendation 3: The resolution of the figure needs to be improved.

Response 3: Thank you very much for pointing out this issue. We have split out panels A and B into two separate figures (figures 1 and 2) in the revised manuscript and improved the resolution of these figures in the revised manuscript (page10-11 of 26).  

Question & Recommendation 4: It is better to mention cell lines used for checking the effect of the drug. For example, in the case of Cixutumunab (Line number 382-384), the author mentions “It was shown to promote the internalization of IGF-1R and induce cell cycle arrest and 383 apoptosis in both androgen-dependent and androgen-independent PCa cells”. It is better if the cell lines used in the study are mentioned wherever necessary.

Response 4: Thank you very much for offering an appropriate proposal. We have added the name of cell lines mentioned in this manuscript to the suitable position according to your suggestion. The main cell lines were list as follows: ARCaPE and ARCaPM、LNCaP、C4-2B、PC-3 and DU-145 cell lines ( Reference 46), PNT2 and DU145 cell lines (Reference 68), LuCaP 35 and LuCaP 35 V (Reference 97), LNCaP and 22Rv1 cells lines (Reference 105, Reference 107).

Sincerely,

Feng Pan, M.D./Ph.D.
Department of Urology
Union Hospital, Tongji Medical College
Huazhong University of Science & Technology
1277 Jiefang Avenue
Wuhan  430022
China
